# The STING-IFN-β-Dependent Axis Is Markedly Low in Patients with Relapsing-Remitting Multiple Sclerosis

**DOI:** 10.3390/ijms21239249

**Published:** 2020-12-04

**Authors:** Lars Masanneck, Susann Eichler, Anna Vogelsang, Melanie Korsen, Heinz Wiendl, Thomas Budde, Sven G. Meuth

**Affiliations:** 1Department of Neurology with Institute of Translational Neurology, University Hospital Münster, 48149 Münster, Germany; lars.masanneck@uni-muenster.de (L.M.); anna.vogelsang@ukmuenster.de (A.V.); heinz.wiendl@ukmuenster.de (H.W.); meuth@uni-duesseldorf.de (S.G.M.); 2Department of Neurology, Heinrich Heine University Düsseldorf, 40225 Düsseldorf, Germany; melanie.korsen@med.uni-duesseldorf.de; 3Institute of Physiology I, University of Münster, 48149 Münster, Germany; tbudde@uni-muenster.de

**Keywords:** STING, cGAMP, interferon-beta, multiple sclerosis, RRMS, EAE

## Abstract

Cyclic GMP-AMP-synthase is a sensor of endogenous nucleic acids, which subsequently elicits a stimulator of interferon genes (STING)-dependent type I interferon (IFN) response defending us against viruses and other intracellular pathogens. This pathway can drive pathological inflammation, as documented for type I interferonopathies. In contrast, specific STING activation and subsequent IFN-β release have shown beneficial effects on experimental autoimmune encephalomyelitis (EAE) as a model for multiple sclerosis (MS). Although less severe cases of relapse-remitting MS (RRMS) are treated with IFN-β, there is little information correlating aberrant type I IFN signaling and the pathologic conditions of MS. We hypothesized that there is a link between STING activation and the endogenous production of IFN-β during neuroinflammation. Gene expression analysis in EAE mice showed that *Sting* level decreased in the peripheral lymphoid tissue, while its level increased within the central nervous system over the course of the disease. Similar patterns could be verified in peripheral immune cells during the acute phases of RRMS in comparison to remitting phases and appropriately matched healthy controls. Our study is the first to provide evidence that the STING/IFN-β-axis is downregulated in RRMS patients, meriting further intensified research to understand its role in the pathophysiology of MS and potential translational applications.

## 1. Introduction

Multiple sclerosis (MS) is a chronic demyelinating disease of the central nervous system (CNS) characterized by an interplay of inflammatory and neurodegenerative processes resulting from aberrant CNS-directed immune responses [1]. The currently available therapy options—with distinct immunomodulatory or immunosuppressive effects—mainly reduce the frequency and severity of relapses but cannot cure the disease [2]. Among the various treatments, the first available disease-modifying therapy for treating MS was interferon (IFN)-β, which was approved for Europe in 1998 after a series of successful trials [3,4,5,6]. Today IFN-β is still applied as a first-line treatment for less severe cases of relapse-remitting MS (RRMS) [7,8]. While its exact mode of action is not fully understood, the beneficial effects of IFN-β seem to be mediated by targeting innate as well as adaptive immune cells [9,10,11,12,13].

Physiologically, IFN-β is a naturally occurring cytokine of the human immune system. It is produced by various kinds of cells, including myeloid cells like bone marrow-derived cells (BMDC), upon recognition of pathogenic components such as viral nucleic acids [14,15,16]. Bacterial infections can also trigger the secretion of IFN-β [15,17]. Once released as type I IFN, it specifically binds to the IFN α/β receptor (IFNAR) on its target cells, thereby initiating pleiotropic effects that trigger enhanced immune responses to combat infections [14,18,19]. Interestingly, the peripheral activity of IFN-β is reduced in RRMS patients [20]. Furthermore, a low IFN-β signature in myeloid antigen-presenting cells (APC) seems to predict a better response to IFN-β therapy [21]. However, the precise mechanisms and actions leading to an aberrant type I IFN response under the pathologic conditions of MS are not yet understood.

Novel findings in the last decade revealed that the induction of IFN-β is triggered, among others, by the stimulator of IFN genes (STING) pathway [15,17,22,23]. This pathway is initiated by cytosolic sensors upon recognition of pathogen-associated molecular patterns (PAMP), like virus-derived nucleic acids. As the most relevant initiator of this pathway within the innate immune system in humans, cyclic GMP-AMP (cGAMP) synthase (cGAS) detects cytosolic nucleic acid and subsequently produces 2′3′-cGAMP, a cyclic dinucleotide (CDN) [15,17,22,23,24,25,26]. In turn, 2′3′-cGAMP serves as a second messenger and can directly bind to the endoplasmic reticulum (ER)-resident adapter protein STING, thus inducing the TANK-binding kinase 1/interferon regulatory factor-3-dependent IFN-β production [22,27]. In the case of the inappropriate recognition of self-derived endogenous nucleic acid, there is a risk of triggering autoimmune and autoinflammatory diseases, as already assumed for type I interferonopathies [28,29]. Several lines of evidence suggest a critical role of the STING pathway in MS. Insights from experimental autoimmune encephalomyelitis (EAE) studies, a commonly used animal model to investigate molecular and cellular mechanisms underlying the immunopathogenic processes of MS, proved the beneficial effects of the specific activation of STING in vivo. One study demonstrated that administration of STING-activating reagents, i.e., DNA nanoparticles (DNP) and a bacterial CDN, significantly reduces disease severity along with delaying EAE onset due to a type I IFN-dependent production of indoleamine 2,3 dioxygenase (IDO), a natural immunoregulatory enzyme, in dendritic cells [30]. Lemos et al. further demonstrated that the beneficial activity of IDO, which can suppress inflammatory T cell responses, is fostered by manipulating STING/IFN-I-dependent signaling in EAE mice [31]. In this context, the importance of the cytosolic sensor cGAS was also emphasized [31]. Together with the finding that experimental blockage with the antiviral drug ganciclovir alleviated EAE by decreasing microglial reactivity in a STING/type I IFN-dependent fashion [32], these data suggest this pathway as a novel target for treating MS.

A causal link between STING activation and the endogenous production of IFN-β in MS has not yet been analyzed. In the present study, the regulation of the STING/IFN-β pathway during neuroinflammation regarding molecular mechanisms of IFN-β production and its regulation in murine and human cells was explored. To our knowledge, we are the first to report results that indicate the downregulation of the cGAS-STING/IFN-β-axis in immune cells of patients suffering from RRMS.

## 2. Results

### 2.1. Sting Expression Rises in CNS Tissue during Neuroinflammation

To gain deeper insights into the complexity of the STING/IFN-β-axis (depicted in Appendix A) during neuroinflammation, various immune or CNS resident cell types associated with the immunopathology of EAE and MS were initially analyzed by quantitative real-time polymerase chain reaction (RT-qPCR) for their transcript levels of *Tmem173*, encoding STING. Our examinations demonstrated that, along with BMDC and mouse brain microvascular endothelial cells (MBMEC), CD4^+^ and CD8^+^ T lymphocytes, as well as regulatory thymus-derived CD4^+^CD25^high^ T (tT_reg_) cells and their non-regulatory counterparts, exhibit the most augmented expression of *Tmem173* under naïve conditions compared to B cells. These data are in line with previously reported data showing that the protein level of STING on murine T lymphocytes is comparable to that on macrophages [33]. Moreover, CNS tissue isolated from naïve C57BL/6J mice showed only a marginal expression of *Tmem173* (Figure 1A).

Next, we actively induced chronic EAE in C57BL/6J mice to determine the involvement of STING during neuroinflammation in vivo and found a trend for decreasing *Tmem173* expression over the disease course in the peripheral lymphoid tissue (Figure 1B), where T cells are activated before infiltrating the CNS [34]. Interestingly, in contrast to naïve conditions, the expression of *Tmem173* is distinctly increased in the inflamed spinal cord parenchyma. This elevation at the main site of inflammatory lesions in chronic EAE [34] seems to correlate with disease progression since the highest *Tmem173* expression was detected at disease maximum. Its level slightly decreased during the chronic EAE phase but was still high in comparison to the gene expression level in naïve spinal cord parenchyma (Figure 1C).

### 2.2. Human Myeloid Cells Exhibit a Pronounced Expression of STING

Based on the above results, we next focused on the STING/IFN-β-axis in humans. The relatively high expression of *Tmem173* in murine T cells and myeloid BMDC prompted us to investigate its expression levels in humans. Also, we analyzed CD4^+^ T cells expressing the immunotolerizing molecule human leukocyte antigen G (HLA-G) since these cells describing another potent tT_reg_ cell subset that seem to be relevant during immunoregulation in the pathogenesis of MS, as recently described by our group [35,36,37]. RT-qPCR analysis of peripheral blood mononuclear cells (PBMC)-derived subsets from healthy donors (HD) revealed that myeloid cells (marked by the expression of CD33) showed significantly higher *TMEM173* expression with a ΔC_t_ of 14.52 ± 0.14 compared to the analyzed subpopulations of T lymphocytes (Figure 2A).

STING analysis on the protein level of human PBMC by multi-color flow cytometry exhibited similar patterns compared to their isotype control (Figure 2B). Our analysis, with 0.19 ± 0.30%, revealed almost no STING expression for CD19 expressing B cells, which is in accordance with the literature, and utilizing western blot analysis shows that primary human B cells are deficient for STING protein expression [38] (Figure 2B—left panel). In contrast, CD3 expressing lymphocytes showed a marginal expression of STING, with 6.86 ± 2.69%, (Figure 2B—middle panel), while myeloid cells, here marked by the expression of CD11b, had the highest protein expression of STING, which is in line with their transcript level data given above. 45.15 ± 8.95% of CD11b expressing cells expressed STING within the human PBMC population under naïve conditions (Figure 2B—right panel). In conclusion, murine and human immune cells slightly differ in their expression levels of STING.

### 2.3. Myeloid Cells Show a Strong Type I IFN Response upon STING Activation with 2′3′-cGAMP

To further elucidate the functional role of the STING/IFN-β-axis for human immune cells, we used CDN 2′3′-cGAMP co-delivered with the permeabilizing agent digitonin to treat PBMC of HD to activate the ER-resident adapter protein STING (Appendix A). After 4 h of treatment, a type I IFN response could be measured by quantification of the expression of *IFNB1* and *IFNA2* genes, encoding IFN-β and IFN-α, respectively. Both the expression of *IFNB1* (fold change of 210.4 ± 41.1) and *IFNA2* (fold change of 71.2 ± 13.6) were considerably increased in samples treated with 2′3′-cGAMP plus digitonin compared to controls receiving only one of the two (Figure 3A,B). In line with the latter results, protein levels with conclusive evidence for IFN-β and IFN-α could be verified after 24 h solely within the obtained culture supernatants of samples treated with 2′3′-cGAMP in the presence of digitonin (Figure 3C,D).

Given the strong type I IFN response after the specific activation of human PBMC with 2′3′-cGAMP, we analyzed the functional impact on CD33 expressing myeloid cells, which showed the highest *TMEM173* expression within this study. Digitonin-based 2′3′-cGAMP stimulation of CD33^+^ myeloid cells led to a markedly stronger IFN-β response on the mRNA level than the unsorted PBMC shown in Figure 3A. CD33^+^ cells exhibited an *IFNB1* fold change of 1597 ± 478 (Figure 4A). Interestingly, a negative fold change of −78.82 ± 7.14 could be detected for the expression of *TMEM173* after activation (Figure 4B), indicating a feedback downregulation of STING. On the contrary, an upregulation of *MB21D1* transcripts encoding the pathway’s initiating enzyme cGAS was found upon specific CDN-triggered STING activation (Figure 4C).

### 2.4. PBMC of MS Patients Exhibit Lower Expression of the STING/IFN-β-axis

To explore the role of the STING/IFN-β pathway in MS, we compared the gene expression levels of the central players in PBMC of HD with those obtained from naïve RRMS patients in either relapse or remission. In detail, PBMC samples of both MS groups were assessed regarding gene expression patterns for *MB21D1*, *TMEM173*, *IFNB1*, *IFNA2,* and *IFNAR1*, and are displayed in Figure 5A–E as ΔΔC_t_ relative to the HD control group [39]. The gene expression results are also given as fold change values in Appendix A.

During neuroinflammation, principal genes involved in the STING/IFN-β pathway seem to be considerably downregulated in MS patient-derived PBMC compared with their expression in HD. The enzyme cGAS, which initiates the pathway, is downregulated during the relapse phase of MS, with significantly lower amounts of *MB21D1* transcripts (ΔΔC_t_ 3.18 ± 0.62 relapse group) compared to the remission group (ΔΔC_t_ 0.70 ± 0.34) (Figure 5A). A slight but significant downregulation could be proven for the transcript levels of both *TMEM173* and *IFNB1* within the MS relapse group. This effect was not observed in the remission group, which showed expression levels close to the HD group for these genes (Figure 5B,C). In contrast, the gene expression levels of *IFNA2* were significantly upregulated within PBMC of both MS groups (ΔΔC_t_ −2.49 ± 0.32 relapse group; ΔΔC_t_ −3.51 ± 0.57 remission group) compared to the HD group (ΔΔC_t_ 0.00 ± 0.86) (Figure 5D). Interestingly, the remission group had a significantly higher expression of the type I IFN receptor gene *IFNAR1* (ΔΔC_t_ −2.42 ± 0.67) compared to both the HD (ΔΔC_t_ 0.00 ± 0.70) and the relapse group (ΔΔC_t_ 0.43 ± 1.01) (Figure 5E). Together, these data suggest that the STING/IFN-β pathway is significantly altered in the peripheral blood cells of patients with RRMS, indicating a vital immunomodulatory role of this pathway, not only in mice but also in humans.

## 3. Discussion

The potential role of the STING/IFN-β-signaling pathway in mechanisms of neuroinflammation has recently been brought to the center of attention [30,31,32]. However, the effects and functions in both general (neuro) inflammatory processes and the more specific MS-related circumstances remain unclear. Our present study gives novel insights into the role of the STING/IFN-β-signaling pathway in the pathophysiological context of both EAE and RRMS. Our findings indicate some relevant interrelationships that merit further research efforts. We made the striking observation that over the course of chronic EAE, the level of *Sting* expression decreased in the peripheral lymphoid tissue, while its level increased at the main site of inflammatory lesions, the inflamed spinal cord parenchyma. Maximal clinical signs of EAE are associated with extensive immune cell infiltration mainly consisting of macrophages and T cells into the CNS through a disrupted blood-brain barrier [34]. Given that naïve CNS tissue barely expressed the gene, leukocyte migration is a likely source for elevated *Sting* expression within the CNS of EAE induced mice. In line with our data, Mathur et al. showed detectable levels of STING protein expression in immunohistochemistry staining of the cerebellum of EAE mice at disease maximum, whereas no detectable levels were found under naïve conditions [32]. In accordance with our observations gained from an animal model of MS, the level of *STING* expression was downregulated in peripheral immune cells during the acute phases of RRMS in comparison to clinically stable RRMS patients and appropriately matched HD. Moreover, the expression of further central players of the STING/IFN-β-signaling pathway, like the initiator *cGAS* and *IFN-β* as a critical final product, was decreased in RRMS patients.

Infection with viruses, protozoa, or bacteria leads to an accumulation of cytosolic nucleic acids of extraneous origin that signals the presence of pathogens to the immune system. The human organism has several mechanisms to defend itself against the potential danger of nucleic acids of aberrant subcellular localization derived from endogenous sources or invading pathogens (e.g., viral infections). The STING/IFN-β-signaling pathway represents a central mechanism that is geared to respond to PAMPs by initiating a type I IFN immune response [15,22,23,24,25,26,27]. Its inappropriate activation has been linked to a variety of autoinflammatory and autoimmune disorders, termed type I interferonopathies. These include, for example, the severe Aicardi-Goutières syndrome, which is associated with a cerebral overproduction of IFN-α [28,29], thus underlining the need for tight regulation of responses. The pathophysiology of MS is also associated with an aberrant type I IFN response since treatment-naïve RRMS patients exhibit not only a significantly reduced endogenous activity of peripheral IFN-β [20] but also a markedly low expression of IFN-stimulated genes [40,41]. The therapeutic administration of exogenous IFN-β, which is commonly used for less severe cases of RRMS, seems to correct this dysregulation, considering that it ameliorates disease activity in MS patients [7,8,41,42]. Together with the knowledge that the STING pathway is critical for the production of IFN-β [15,17,22,23,26,27], and its direct activation showed promising results in the EAE model [30,31,32], it is likely that the STING/IFN-β-signaling pathway could account for the lower endogenous expression levels of IFN-β and IFN-dependent genes underlying the pathophysiology of MS, as described by others [20,40,41]. In line, we further observed that peripheral levels of *Sting* expression were decreased during EAE and in therapy naïve active RRMS patients. These patients showed decreased levels of *cGAS* as an initiator of STING/IFN-β-signaling and IFN-β, while therapy-naïve stable RRMS patients revealed almost the same levels as found in HD.

The effectiveness of IFN-β therapy is attributed to diverse immune regulatory mechanisms [9,10,11,12,13]. Clinical practice, however, has demonstrated restricted potency, as not all patients respond to IFN-β treatment. The characterization of (partial) non-responders and responders to IFN-β therapy disclosed that the presence of neutralizing antibodies to IFN-β could only partly explain the limited response to IFN-β [43]. Also, low type I IFN baseline levels in peripheral myeloid cells could reliably predict an effective therapeutic response to IFN-β, which would ideally be determined before starting treatment [21]. In other words, MS patients with poor response to IFN-β seem to exhibit an already fully activated type I IFN signaling pathway that is refractory to the therapeutic administration of exogenous IFN-β since it cannot be activated further. This was observed for myeloid cells but not for T and B cells [21]. The roles of T and B cells in the adaptive immune response are well known in MS pathophysiology. Although the importance of myeloid cells, including monocytes, macrophages, microglia, and dendritic cells, as part of the innate immune system is recognized in this context, their exact functions are less commonly considered in MS. Interestingly, we observed that human myeloid cells show not only the highest gene and protein expression of STING in comparison to T and B cells but also an elevated expression of IFN-β after direct STING activation by 2′3′-cGAMP—a finding that suggests a possible relationship between STING activation and endogenous production of IFN-β during neuroinflammation. We also speculate that the limited therapeutic effect of IFN-β in some MS patients may be attributed to a STING/IFN-β-signaling pathway imbalance in myeloid cells. However, this needs to be clinically proven in further experiments since this study was not designed to characterize (partial) non-responders and responders to IFN-β treatment in terms of their STING expression patterns within the various myeloid-derived immune cells. Here, we included defined groups of RRMS patients—an approach that we consider as an advantage in comparison to studies with mixed clinical MS groups.

Another point to consider is the methodology of CDN stimulation itself. Even though digitonin permeabilization with subsequent CDN stimulation is widely applied [17,23,32,38], potential off-target effects cannot be excluded. Therefore, we standardly incubated the cells with just the permeabilization buffer or 2′3′-cGAMP, both as additional controls, and did not observe any strong influence on the effects of CDN stimulation, as marked by no or low expression of IFN-α and IFN-β. Further, we did not have the opportunity to measure the amount of 2′3′-cGAMP reaching the cytoplasm for activation of the STING/IFN-β-signaling pathway. A radioactive-labeled 2′3′-cGAMP represents a way to monitor the location and distribution of CDN during stimulation, similar to the method used by Burdette et al. in HEK293T cells [23], but was not a feasible approach to be applied within our laboratory setting. Since we adapted their general CDN stimulation protocol, we assumed that the intracellular concentrations reached similar levels. Of note, to the best of our knowledge, this is the first time that the protein expression of STING was intracellularly assessed using flow cytometric analysis, which paves the way for future research. In accordance with our findings on the mRNA levels, human myeloid cells comprise the subset with the strongest STING protein signal determined via flow cytometry, indicating a reliable method to quantify its protein level.

Given the predominant role of the STING/IFN-β-signaling pathway in antiviral immune response, it was of interest to investigate an indispensable virus detection system in the context of neuroinflammation, as viruses are being considered as etiological factors in MS. The role of the Epstein-Barr virus (EBV), which is a natural stimulus leading to the production of type I IFN, is often discussed in relation to MS. It is currently argued whether EBV-infected B cells may also contribute to triggering aberrant immune responses in MS [44,45]. It is reasonable to assume that the EBV can exploit human B lymphocytes as a reservoir for persistent infection because these cells are lacking detectable levels of the central adaptor protein STING, necessary to activate IFN I secretion upon the detection of foreign viral nucleic acids [38]. Nevertheless, it is unlikely that an EBV imbalance accounts for the different type I IFN responses in IFN-β responders and non-responders of MS patients, as both groups showed similar levels of EBV reactivation [21]. Concerning the current coronavirus pandemic, recent functional in vitro analysis found that IFN-β treatment effectively blocks SARS-CoV-2 replication in a dose-dependent manner, indicating a possible side benefit for MS patients treated with IFN-β [46].

An aberrant type I IFN response might, on the one hand, predict the course in MS [20,40] and, on the other hand, the response to IFN-β therapy [21]. However, the precise mechanisms and actions leading to an aberrant type I IFN response under the pathologic conditions of MS are not yet understood. Overall, we suggest that the STING/IFN-β-signaling may be the cause of the aberrant type I IFN response during neuroinflammation, thus playing a vital immunomodulatory role not only in mice but also in humans. A better understanding of its immunomodulatory role could lead to a more profound assessment of IFN-β therapy responders and give insights into the pathophysiology of signaling events underlying MS in general. Moreover, the recent discovery that the STING/IFN-β-signaling pathway is linked to the pathology of EAE and possibly MS is strengthened by the fact that its specific activation leads to an improved clinical EAE course by inducing an IFN type I response [30,31,32]. In this respect, new highly potent CDN STING agonists may be of future interest [30,31,32]. Considering these different perspectives, targeting the STING/IFN-β-signaling is of therapeutic interest [47]. Since cGAS is up-regulated and mediates inflammation in the brain in neurodegenerative disorders like Huntington’s disease, central STING/type I IFN-dependent effects may also be relevant [48]. This takes into account that the antiviral drug ganciclovir alleviated EAE by decreasing microglial reactivity in a STING/type I IFN-dependent fashion [32] and that HIV-infected patients showed a lower MS incidence when treated with antiviral drugs [49,50]. Notably, since pacemaker channels present in the thalamus show a unique nanomolar affinity for 2′3′-cGAMP, there is an interesting possibility that these ion channels may contribute to central MS pathology [51,52,53,54,55]. However, we need more detailed information from further studies to draw definite conclusions regarding the role of the STING/IFN-β-signaling in the disturbed homeostasis of MS and to verify our observations.

## 4. Materials and Methods

### 4.1. Cell Preparation and Isolation

Murine CD4^+^ and CD8^+^ T cells, CD4^+^CD25^high^ tT_reg_ cells and their respective non-regulatory counterparts (CD4^+^CD25^-^ T cells), and B cells were isolated from spleen single-cell suspensions of naïve C57BL/6J mice (Charles River Laboratories, Sulzfeld, Germany) using magnetic-activated cell sorting (MACS) kits according to the manufacturer’s instructions (Miltenyi Biotec, Bergisch Gladbach, Germany). The isolation of MBMEC was performed as previously described [56]. In brief, MBMEC were prepared from the cortex of naïve C57BL/6J mice and seeded onto collagen IV/fibronectin-coated 24-well dishes for 5 days before collection. BMDC were generated as described by Lutz et al. [57]. Briefly, bone marrow was flushed out of the dissected femur of naïve C57BL/6J mice and afterwards washed three times with BMDC medium (10 mM HEPES, 25 μM 2-mercaptoethanol, 10% fetal calf serum, 1% penicillin/streptomycin) in RPMI-1640 medium (ThermoFisher Scientific, Waltham, MA, USA). The cells were then cultured in the presence of 10% granulocyte-macrophage colony-stimulating factor for a total of 10 days according to the protocol.

Human CD4^+^ and CD8^+^ T cells were purified from LRS (leucocyte reduction system) chamber content (Department of Transfusion Medicine, University Hospital Münster, Germany) of anonymous HD by negative selection using the respective RosetteSep enrichment kits (StemCell Technologies, Vancouver, BC, Canada). Purified untouched CD4^+^ T cells were further fractionated by MACS (Miltenyi Biotech) for CD4^+^CD25^high^ or CD4^+^HLA-G^+^ tT_reg_ cells as described previously [58]. CD4^+^ T cells depleted of HLA-G and CD25 (here defined as CD4^+^CD25^−^ T cells) were also used for further procedures. Human CD33^+^ cells were isolated using the respective MACS kit according to the manufacturer’s protocol (Miltenyi Biotec). PBMC were obtained from LRS chambers by density gradient centrifugation using Lymphoprep separation medium (BioLegend, San Diego, CA, USA). After isolation, lysis of red blood cells (RBC) was performed with RBC lysis buffer (BioLegend) according to the manufacturer’s instructions. Purity after cell isolations was routinely checked using flow cytometry and was higher than 90% in all cases (data not shown).

### 4.2. Murine Tissue Preparation

Spinal cords and brains were obtained from naïve C57BL/6J mice that were transcardially perfused with Dulbecco’s phosphate-buffered saline (DPBS; Merck KGaA, Darmstadt, Germany) for RNA isolation. Spinal cords were also harvested from EAE induced mice after transcardial perfusion on days 16 and 30 after immunization. Active EAE was induced by immunizing 8–12-week-old female mice with 200 μg murine myelin oligodendrocyte glycoprotein peptide fragment 35–55 (sequence: MEVGWYRSPFSRVVHLYRNGK; purity > 99% measured by high-performance liquid chromatography; Charité, University Hospital Berlin, Berlin, Germany) according to our previously published protocol [59]. Organs of mice were removed in strict accordance with the laws and regulations for animal care and scientific use of the regulatory authorities in North Rhine-Westphalia, Germany (84-02.04.2013.A142 approved 10/2013, 84-02.05.50.17.019 approved 10/2017).

### 4.3. Cell Cultures

We used a digitonin based permeabilization method with minor modifications to treat human PBMC with 2′3′-cGAMP [17]. Thus, 3 × 10^7^ freshly isolated PBMC were resuspended in 2.1 mL of freshly prepared permeabilization buffer (50 mM HEPES, 100 mM KCL, 3 mM MgCl_2_, 0.1 mM 1,4-dithiothreitol, 85 mM Sucrose, 0.2% bovine serum albumin (BSA), 1 mM adenosine 5′-triphosphate disodium, 0.1 mM guanosine 5′-triphosphate sodium, 3 μg/mL digitonin and 10 μg/mL 2′3′-cGAMP (InvivoGen, San Diego, CA, USA), pH = 7.0) in dH_2_0, here defined as cGAMP + D group, and incubated at 37 °C and 5% CO_2_ for 10 min in the dark. As a control, PBMC were either treated with permeabilization buffer without 2′3′-cGAMP (here defined as D group) or 10 μg/mL 2′3′-cGAMP in permeabilization buffer without 3 μg/mL digitonin (here defined as cGAMP group). After centrifugation at room temperature with 450 g for 5 min, PBMC were resuspended in 1 mL of X-VIVO 15 medium (Lonza, Verviers, Belgium) and seeded in a 24-well plate for 4 h or 24 h at 37 °C and 5% CO_2_. After incubation, the cells were centrifuged after 24 h at 300 g for 10 min to obtain cell culture supernatants for further procedures.

### 4.4. Flow Cytometry

Before staining, Fc receptors were blocked according to the manufacturer’s instructions (Miltenyi Biotec). Cell surface staining using the antibodies CD3 (clone: HIT3a, BioLegend), CD11b (clone: M1/70 BioLegend) and CD19 (clone: HIB19, Beckman Coulter, Krefeld, Germany) was carried out in DPBS buffer with 0.1% BSA and 0.1% sodium azide for 30 min at 4 °C. The intracellular antigen STING (1:10; clone: T3-680, Becton, Dickinson and Company (BD), Franklin Lakes, NJ, USA) or its corresponding isotype control (mouse IgG1, κ) were stained using the Cytofix/Cytoperm fixation/permeabilization kit according to the manufacturer’s instructions at RT instead of 4 °C (BD). Finally, stained samples were assayed on a multi-color flow cytometer (Gallios, Beckman Coulter) using Kaluza software v2.1 (Beckman Coulter). Fixable viability dye eFluor 780 (1:10000; ThermoFisher Scientific) was used for dead cell exclusion. Moreover, for all analyses, cell doublets were excluded to ensure single cell counting.

### 4.5. Immunoassay

PBMC culture supernatants were analyzed for their IFN concentrations using the LEGENDplex type 1/2/3 IFN panel (sensitivity: 1.74 pg/mL for IFN-α-2a, 3.19 pg/mL for IFN-β, BioLegend) according to the manufacturer’s instructions. For calculations and further analysis, the LEGENDplex Software v.8.0 (BioLegend) was used.

### 4.6. Study Subjects

Thirty human subjects, including healthy individuals and clinically defined MS patients diagnosed according to the 2017 revised McDonald diagnostic criteria [60], were recruited to the study. All human subjects gave informed consent for study participation in accordance with the Declaration of Helsinki, and the study was approved by the Ethics Committee of the University of Münster (2016-053-f-S). Fresh blood samples were obtained from 20 patients with confirmed RRMS diagnosis (14 females, 6 males) who had been referred to the Department of Neurology with Institute of Translational Neurology, University Hospital Münster. The MS patients were either in relapse (mean age: 36.20 ± 4.47; n = 10) or remission (mean age: 31.80 ± 2.89; n = 10). All patients included in the study had not received any immunomodulatory treatment except corticosteroids, with the last dose administered at least 3 months prior to study entry. In parallel, 10 HD (5 females, 5 males) with a mean age of 42.33 ± 4.87 were included in the study. Fresh blood was collected in EDTA-containing tubes (K2E Vacutainer, BD) for the isolation of PBMC and frozen in aliquots at −80 °C for 48 h before being transferred to a liquid nitrogen tank at −196 °C.

### 4.7. RNA Isolation

Samples of PBMC were thawed according to the standard operating procedure of our clinic [61], and murine tissue samples were homogenized before RNA extraction. Total RNA was extracted from both using TRIzol (ThermoFisher Scientific) according to the manufacturer’s instructions.

### 4.8. Quantitative Real-Time Polymerase Chain Reaction

Synthesis of cDNA was performed with Maxima reverse transcriptase and used as a template in the subsequent RT-qPCR using Maxima probe/ROX qPCR mastermix according to the manufacturer’s recommendations on an Applied Biosystems Step One Plus RT-qPCR instrument (ThermoFisher Scientific). The following TaqMan gene expression assays were used (all purchased from ThermoFisher Scientific): *Tmem173* (Mm01158117_m1), *TMEM173* (Hs00736956_m1), *MB21D1* (Hs00403553_m1), *IFNB1* (Hs01077958_s1), *IFNA2* (Hs00265051_s1), *IFNAR1* (Hs01066116_m1), and eukaryotic 18S ribosomal RNA (rRNA; 4319413E). Results were analyzed using the StepOne software v2.3 (ThermoFisher Scientific) and the comparative C_t_ method (C_t_ is defined as the cycle threshold). Data are expressed as ΔC_t_ (defined as C_t_ gene of interest—C_t_ housekeeping gene) or ΔΔC_t_ relative to a control group (defined as ΔC_t_ of sample—ΔC_t_ of control group) [39]. The genes of interest were normalized to the housekeeping gene (18S rRNA). All data points were measured in triplicates, and mean C_t_ values were used for the calculations. If suited, the corresponding x-fold change relative to control was depicted and calculated using the common 2^−ΔΔCT^ transformation [39]. Therefore, the mean corresponding ΔΔC_t_ value of the control group was normalized to 1 and set as the baseline (BL), thus serving as a reference.

### 4.9. Statistical Analysis

For each type of experiment, group sizes are given in the figure legends. Data were presented as the mean ± standard error of the mean (SEM) from at least three independent experiments. In the case of multiple comparisons, a one-way analysis of variance (ANOVA) and Kruskal–Wallis test with Dunn post hoc analysis was used. Mann–Whitney *U*-test for parametric data without normality datasets was used. D’Agostino-Pearson omnibus was used to assess the normality of a dataset. Data were analyzed using Prism 5.04 (GraphPad Software, San Diego, CA, USA), and values of probability (*p*) < 0.05 were considered as statistically significant. The level of significance was labeled as NS (not significant), * *p* < 0.05, ** *p* < 0.01 or *** *p* < 0.001.

## Figures and Tables

**Figure 1 ijms-21-09249-f001:**
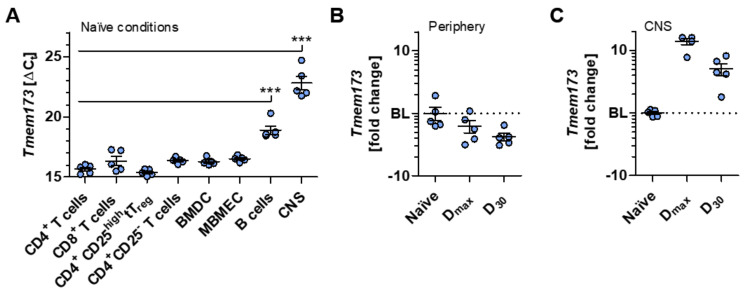
*Sting* is upregulated in central nervous system (CNS) tissue under neuroinflammatory conditions. (**A**) Quantitative real-time polymerase chain reaction (RT-qPCR) gene expression analysis of *Tmem173* encoding the stimulator of interferon genes (STING) was performed in different immune or CNS resident cell types. ∆C_t_ values are illustrated for the indicated cell subsets or tissue of naïve C57BL/6J mice. (**B**) Fold change analysis of *Tmem173* in peripheral lymphoid tissue of active experimental autoimmune encephalomyelitis (EAE)-induced C57BL/6J mice is shown for days 16 (D_max_) and 30 (D_30_) post-immunization relative to naïve tissue. (**C**) Fold change of *Tmem173* was calculated from thoracic spinal cords of EAE-induced C57BL/6J mice at D_max_ and D_30_ post-immunization relative to naïve tissue. In (**B**,**C**), the mean corresponding ΔΔC_t_ value of the control group was normalized to 1 and set as the baseline (BL; displayed as dotted line), thus serving as a reference. Data in (**A**–**C**) are mean ± standard error of the mean averaged from five mice per condition performed with technical triplicates. Data in (**A**) were analyzed by one-way analysis of variance (ANOVA), and the level of significance was labeled as *** probability (*p*) < 0.001. *Abbreviations: Thymus-derived regulatory T cells (tT_reg_), bone marrow-derived cells (BMDC), murine brain microvascular endothelial cells (MBMEC)*.

**Figure 2 ijms-21-09249-f002:**
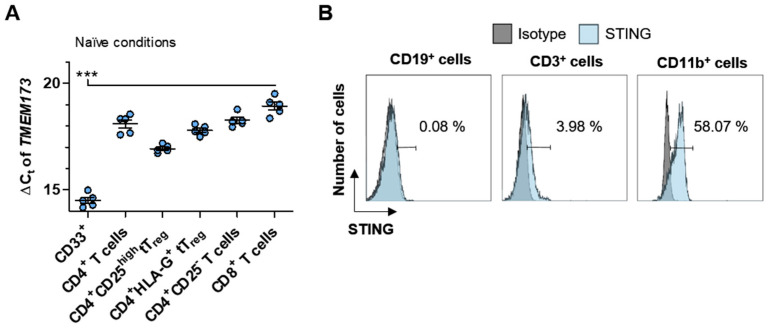
Human myeloid cells show a higher expression of STING than lymphoid cells. (**A**) Illustrated are the expression analysis of *TMEM173* in CD33 expressing myeloid cells and different human T cell subpopulations. All cell populations were isolated from peripheral blood mononuclear cells (PBMC) of healthy donors (HD), and each symbol represents the resulting ΔC_t_ value of an individual HD. (**B**) Percentage of STING protein expression in comparison to its corresponding isotype control is shown for CD19^+^ (left panel), CD3^+^ (middle panel), and CD11b^+^ (right panel) cells as one representative result of 5 independent flow cytometric analyses. Each symbol (**A**) represents an individual HD. Data in (**A**) were analyzed by one-way ANOVA, and the level of significance was labeled as *** *p* < 0.001. *Abbreviations: Human leukocyte antigen G (HLA-G)*.

**Figure 3 ijms-21-09249-f003:**
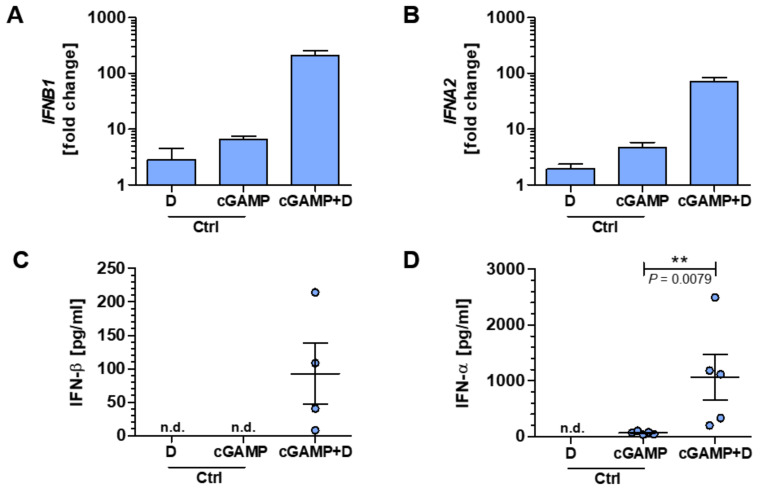
Specific cyclic dinucleotide (CDN) stimulation of PBMC leads to interferon (IFN) type I response. (**A**–**D**) PBMC were stimulated with the CDN 2′3′-cGAMP co-delivered with digitonin (here defined as cGAMP+D group) or solely with one of the two and afterward analyzed for their IFN type I responses. Gene expression analysis of *IFNB1* (**A**) and *IFNA2* (**B**) are shown as fold change relative to an untreated control after 4 h of stimulation. After 24 h of stimulation, the protein concentrations of IFN-β (**C**) or IFN-α (**D**) were determined. Some protein concentrations were under the detection threshold and are indicated as not detectable (n.d.). Of note, in (**C**), one sample was also below the detection threshold within the cGAMP + D group. Each data point represents an individual HD. Data were compiled from five independent experiments. Data in (**D**) were analyzed by Mann-Whitney *U*-test, and the level of significance was labeled as ** *p* < 0.01. *Abbreviations: 2′3′- cyclic guanosine monophosphate–adenosine monophosphate (2′3′-cGAMP), control (Ctrl)*.

**Figure 4 ijms-21-09249-f004:**
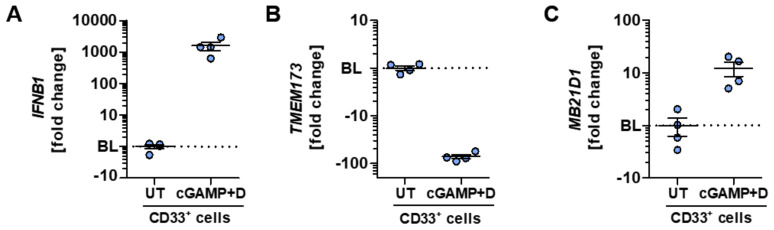
Highly STING-expressing myeloid cells exhibit a strong response to specific CDN stimulation. (**A**–**C**) CD33^+^ myeloid cells of HD were stimulated with 2′3′-cGAMP plus digitonin (here defined as Cgamp + D group) or left untreated (UT) and afterward analyzed for the expression of genes belonging to the STING/IFN-β-axis. Gene expression analysis of *IFNB1* (**A**), *TMEM173* (**B**), or *MB21D1* (**C**) are shown as fold change relative to the untreated control after 4 h of stimulation. In (A–C), the mean corresponding ΔΔC_t_ value of the control group was normalized to 1 and set as the BL (displayed as dotted line), thus serving as a reference. Each data point represents an individual HD. Data were compiled from four independent experiments.

**Figure 5 ijms-21-09249-f005:**
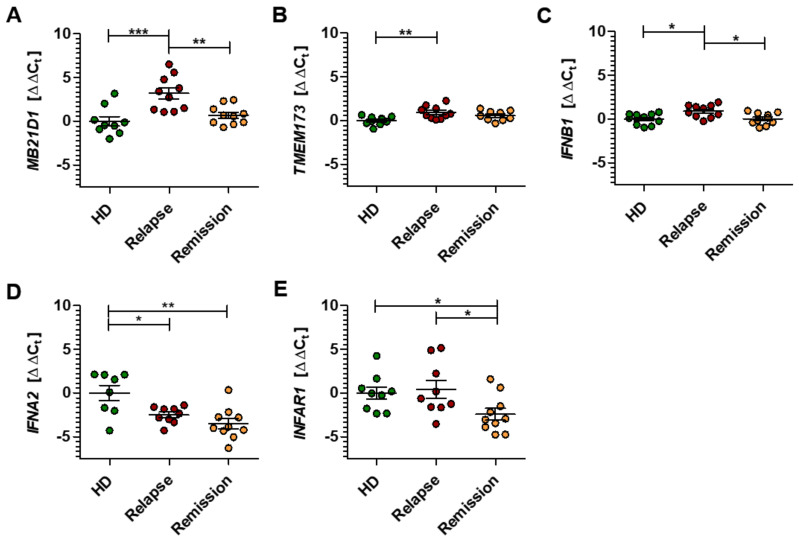
Gene expression pattern of the STING/IFN-β-axis in patients with multiple sclerosis. (**A–E**) The gene expression levels of *MB21D1* (**A**), *TMEM173* (**B**), *IFNB1* (**C**), *IFNA2* (**D**), and *INFAR1* (**E**) were analyzed in PBMC of HD and relapsing-remitting multiple sclerosis patients in relapse or remission. Gene expression is illustrated as ΔΔC_t_ calculated from the corresponding ∆C_t_ values in reference to the control group of HD. Each data point represents an individual HD or patient. Data were analyzed by one-way ANOVA, and the level of significance was labeled as * *p* < 0.05, ** *p* < 0.01 or *** *p* < 0.001.

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
