# Peer review of "The STING-IFN-β-Dependent Axis Is Markedly Low in Patients with Relapsing-Remitting Multiple Sclerosis"

_ijms, 2020, doi:10.3390/ijms21239249_

Round 1
Reviewer 1 Report
The present manuscript hypothesizes a link between STING activation and endogenous production of IFN-beta during neuroinflammation, providing the first evidence of a downregulation of STING/IFN-beta axis in RRMS patients.
I honestly have to declare that the authors clearly expose and define the experimental hypothesis. Moreover, the experimental design is well organized and the results are well linked among them.
I think this manuscript is directly suitable for publication.
Reviewer 2 Report
The authors describe expression of STING/IFNb pathway during EAE in mice and in peripheral immune cells during acute RRMS. MS patients had significant impairment in this pathway. This is the first description of this interesting finding.
A bit more discussion of why the authors selected to examine HLA-G would be beneficial and also the inclusion of results from MS patients in figure 2. Figures 1-4 need the statistical significance denoted.
Minor edits:
Line 224
Disease maximum of EAE is associated with extensive immune cell infiltration mainly of macrophages and T cells into the CNS through a disrupted blood-brain barrier. Given that naïve CNS tissue barely expressed the gene, leukocyte migration is a likely cause for elevated Sting expression within the CNS of EAE induced mice. In line with our data, Mathur et al. solely showed detectable levels of STING protein expression in immunohistochemistry stainings of the cerebellum
Change to:
Maximal clinical signs of EAE are associated with extensive immune cell infiltration, mainly consisting of macrophages and T cells into the CNS through a disrupted blood-brain barrier. Given that naïve CNS tissue barely expressed the gene, leukocyte migration is a likely source for elevated Sting expression within the CNS of EAE induced mice. In line with our data, Mathur et al. solely showed detectable levels of STING protein expression in immunohistochemistry staining of the cerebellum
Round 2
Reviewer 2 Report
Authors have adequately addressed the previous concerns.